# Association Between Ultra-Processed Food Consumption Frequency and Frailty: Findings from the InCHIANTI Study of Aging

**DOI:** 10.3390/geriatrics10050123

**Published:** 2025-09-11

**Authors:** Xin Li, Yichen Jin, Stefania Bandinelli, Luigi Ferrucci, Toshiko Tanaka, Sameera A. Talegawkar

**Affiliations:** 1Department of Exercise and Nutrition Sciences, Milken Institute School of Public Health, The George Washington University, Washington, DC 20037, USA; xli317@email.gwu.edu (X.L.); yjin@email.gwu.edu (Y.J.); 2Geriatric Unit, USL Toscana Centro Firenze, 50122 Florence, Italy; stefania1.bandinelli@uslcentro.toscana.it; 3Translational Gerontology Branch, National Institute on Aging, Baltimore, MD 21224, USA; ferruccilu@grc.nia.nih.gov (L.F.); tanakato@mail.nih.gov (T.T.)

**Keywords:** ultra-processed foods, frailty index, healthy aging, InCHIANTI study

## Abstract

Background/Objectives: As individuals age, they experience declines in multiple physiological domains, which increases their vulnerability to health challenges and frailty. While adherence to healthy dietary patterns has been shown to protect against frailty, consuming ultra-processed foods (UPFs)—which are high in added sugars and saturated fat—may contribute to frailty risk. This study investigates the association between UPF consumption and frailty progression among 938 participants aged 65 years and older who were in the InCHIANTI study, Italy. Methods: The patients’ dietary intakes over the past year were assessed using a validated food frequency questionnaire, with items categorized into food groups based on the Nova classification. Frailty was operationalized using a 42-item frailty index (FI). Multivariable linear regression was used to examine the association between the baseline UPF consumption frequency and baseline frailty status, while linear mixed-effects models were used to examine the frailty progression over time. Results: Overall, the participants with the lowest UPF consumption frequency were younger, had more years of education, and had a lower baseline FI. Higher UPF consumption was significantly associated with a greater baseline FI after adjustments for the sociodemographic and health characteristics (*β* = 0.026, 95% CI = 0.010–0.041, *p* = 0.001), and this difference persisted over a 16.1-year follow-up period (*β* = 0.022, 95% CI = 0.006–0.037, *p* = 0.006). Conclusions: These findings underscore the potential negative health impacts of UPF on frailty prevalence and progression in older adults.

## 1. Introduction

As individuals age, they are more likely to accumulate health deficits and have a higher risk for various chronic diseases and mortality. This concept of age-related health deficits accumulating in older adults and impacting their resiliency to perturbations in health is one of the most used definitions of “frailty” [1], also referred to as the “deficit accumulation model” [2]. While there are various assessment methods for frailty, it is irrefutable that the prevalence of frailty increases with age [3] and frailty is considered a significant barrier to healthy aging since it poses increased risks of hospitalization [4], disability [5], falls [5], and death [6]. However, frailty is not a steady state that inevitably worsens once it sets in, and there is potential to reverse frailty via timely interventions in modifiable factors such as an individual’s diet [7].

Diet, an important modifiable behavioral factor, is associated with several common risk factors of frailty, such as low muscle mass [8], obesity [9], and undernutrition [7]. Observational studies have indicated that higher adherence to healthful dietary patterns is associated with a reduced risk of frailty among older adults across various dietary cultures. In the Nutrition and Health Survey, conducted in Taiwan from 2014 to 2016, individuals categorized into the highest tertile of adherence to a diet rich in fruits, nuts and seeds, tea, vegetables, whole grains, fish, shellfish, and milk were only 4% as likely to be frail as those in the lowest tertile of adherence [10]. In the Framingham Offspring Study in the United States, a one-unit higher Mediterranean-Style Dietary Pattern Score was associated with a reduction in the odds of frailty by 3% in adults older than 60 years [11]. Within the InCHIANTI study, better adherence to the Mediterranean diet (indicated by receiving a Mediterranean Diet Score/MDS more than 6) was associated with a 70% reduction in the odds of developing frailty compared with those with lower adherence (indicated by receiving an MDS less than 3) [12]. One plausible explanation for the observed protective association between these healthful dietary patterns and lower frailty risk that has been discussed in previous studies is the increased intake of healthful and nutrient-dense foods like vegetables and fruits that are rich in phytochemicals with anti-inflammatory properties and the reduced consumption of pro-inflammatory foods such as red and processed meats and ultra-processed foods (UPFs) [7,11,12].

By definition, UPFs are hyper-palatable, affordable, and convenient formulations produced from a series of industrial processes [13]. There has been a consistent rising trend in UPF consumption across the life course worldwide [14,15,16,17]. In the U.S., more than half of the daily energy intake of adults aged 60 years and older comes from foods classified as ultra-processed [17], which raises concerns regarding healthy aging because most ultra-processed foods are energy-dense products that are high in added sugars and saturated fats [18]. Excessive intake of high-fat and high-added sugar products contributes to the development of obesity [19,20], low-grade systemic inflammation [21], increased risks of metabolic conditions like diabetes, and other adverse health outcomes [22], including frailty. In earlier research on frailty, Sandoval-Insauti et al. investigated the association between UPF consumption and incident frailty (ascertained based on Fried’s criteria [23]) using the data from the Study on Nutrition and Cardiovascular Risk Factors, conducted in Spain (Seniors-ENRICA) [24]. The results of this study suggested that higher UPF consumption was strongly associated with frailty risk in adults aged 65 years and above [24]. However, the frailty phenotype utilized in this study is not identical to the frailty index (FI), which is another frequently used operational definition of frailty and an important tool for capturing different dimensions of health in older age [1,2]. Additionally, studies examining UPF consumption and frailty have mainly focused on assessing the risk of frailty at a single time point rather than investigating the trajectories of frailty progression in older adults [24,25,26]. This long-term perspective is crucial for better understanding the overall health and quality of life in an aging population. The current study aims to address this research gap by investigating both the cross-sectional associations between baseline UPF consumption frequency and FI, as well as the longitudinal association between UPF consumption frequency and FI trajectories in older adults from the InCHIANTI study. Based on the findings from the previous studies [24], we hypothesized that a higher baseline consumption frequency of UPF is associated with a higher baseline FI and faster FI progression across the follow-up period, which indicate a higher risk of frailty.

## 2. Methods

### 2.1. Study Design and Population

The InCHIANTI (“Invecchiare in Chianti”) study is a prospective study of aging conducted in two municipalities (Bagno a Ripoli and Greve in Chianti) in Tuscany, Italy. The objective of the InCHIANTI study is to investigate risk factors that impact the walking capability of older adults. Detailed descriptions of the study design have been noted elsewhere [27]. In brief, a total of 1453 participants spanning various age groups were recruited at baseline in 1998–2000 via two-stage sampling methods and then followed longitudinally. Data on study variables were collected using structured home interviews. Full medical and functional examinations were conducted at the study sites every three years, including blood and urine sample collections [27].

In the current analyses, we included 938 participants after excluding those at baseline who were younger than 65 years (*n* = 290), who were missing frailty index (*n* = 24), and who reported implausible energy intakes (≤600 or ≥4000 kcal/day) on the food frequency questionnaire (FFQ) (*n* = 8).

### 2.2. Dietary Assessment and UPF Categorization

The baseline dietary intake data were used to assess each participant’s UPF intake. Participants’ dietary intakes over the previous year were assessed at baseline using an FFQ adapted from the European Prospective Investigation on Cancer and Nutrition study, which was validated for use in the InCHIANTI study [28].

We used the Nova food classification system to categorize foods based on the nature and degree of food processing levels. The rationale behind the Nova food classification system has been explained and detailed elsewhere [13,29,30]. There are four categories within the Nova system: (1) unprocessed or minimally processed foods like fresh fruits and vegetables; (2) processed culinary ingredients like salt, oils, and butter used in cooking; (3) foods that go through basic processing methods like canned vegetables, canned fish, and freshly baked bread; (4) ultra-processed foods that undergo extensive industrial processing with additions of additives for industrially exclusive use—examples are frozen pizzas, chips, and sugary carbonated beverages.

The list of foods in the FFQ used in the InCHIANTI study included 240 items. A four-step process was undertaken to identify foods from the FFQ food list and classify them into the Nova categories. First, all the food items in the original food list were translated from Italian to English before categorization. Second, working independently, three researchers (X.L., Y.J., and S.A.T.) assigned the foods in the food list to the Nova groups based on the nature of processing in these foods. This process followed food group definitions based on the Nova classification system [30], and referenced from studies that were conducted in similar contexts or used similar FFQs [24,25,31]. The food categorization results of the three researchers were triangulated in the third step to finalize the UPF assignments. Food items for which there was consensus on their categorization among all researchers were assigned to the corresponding Nova group. For uncertain food items where a consensus on categorization was not reached, they proceeded to the final step of classification, which was an expert consultation with the PIs of the InCHIANTI study (L.F. and S.B.), who have knowledge of the participants in the cohort and have native familiarity with the dietary habits of the study population. During the categorization process, each food item listed in the FFQ was classified into the appropriate Nova food group based on the nature and extent of processing of the whole product rather than considering its nutrient profile or the actual portion size consumed by participants.

The UPF intake of each participant was calculated by summing the consumption frequency of the UPF items. Energy adjustment of the UPF intake was achieved using the residual method [32]. The analysis used quartiles of the UPF intake frequency residuals as the main predictor.

### 2.3. Operationalization of Frailty Index (FI)

The variable selection and construction of the FI in the InCHIANTI study has been described in detail previously [33]. Briefly, frailty was operationalized using a 42-item FI that comprised major chronic diseases [34] such as hypertension, cancer, and Parkinson’s disease, (instrumental) activities of daily living [35], self-rated health, depressive symptoms (ascertained using the Center for Epidemiologic Studies Depression scale [36]), global cognitive function (assessed by the Mini-Mental State Examination [37]), unintentional weight loss, sedentary behavior, grip strength, and walking speed. The computed FI was a ratio of the sum of these 42 variables representing health deficits to the total number of non-missing components and ranged from 0 to 1, indicating no frailty to severe frailty.

### 2.4. Measurement of Covariates

Sociodemographic characteristics, including age, sex, and years of education, were collected using a structured interview. Self-reported smoking status was categorized into three categories: never smoking, smoking previously, and smoking currently (defined as smoking within 3 years). Body mass index (BMI) of each participant was calculated using weight (kg) and height (m) measured at the study clinic. Since the FI accounts for aspects of physical function such as gait speed and sedentary behavior, baseline physical activity levels were not separately adjusted for in the model to prevent potential over-adjustment.

### 2.5. Statistical Analysis

Differences in baseline characteristics of the cohort participants across UPF consumption frequency quartiles were examined using analysis of variance (ANOVA) for continuous variables and Chi-Square tests for categorical variables. The cross-sectional association between baseline UPF consumption frequency and FI was assessed using multivariable linear regression models. The longitudinal association between baseline UPF consumption frequency and repeated measures of FI scores over follow-up years was examined using linear mixed-effect models, with calculated follow-up years by visit as the random effect of the models. Interaction between UPF consumption frequency and time was tested in the linear mixed-effect models to assess whether the association varied over the follow-up period. For both cross-sectional and longitudinal analyses, the models were adjusted for age, sex, study site, smoking status, BMI, and years of education. The total energy intake of participants was adjusted with the residual method when developing the consumption frequency quartiles. The threshold for statistical significance was set at *p* ≤ 0.05. All analyses were conducted using R version 4.3.2.

## 3. Results

The baseline sociodemographic characteristics of the study participants are presented in Table 1. Among the 938 participants, the mean age (SD) was 74 (6.6) years, with 55.2% being females. No significant differences were observed in the average BMI and smoking status of participants across different UPF consumption frequencies. Compared with those in higher quartiles, participants in the lowest UPF intake quartile (Quartile 1) were more likely to be females, were slightly younger by about two years, and had higher education attainment (*p* < 0.05 for all). Participants categorized as having the highest baseline UPF consumption frequency (Quartile 4) had the highest total daily energy intake and a mean baseline FI of 0.146.

### 3.1. Cross-Sectional Association Between UPF Consumption Frequency and FI at Baseline

In general, older adults with more frequent UPF consumption showed greater frailty at baseline compared to those with the lowest intake, even after adjustments for sociodemographic and health-related factors (Table 2). Significant differences were observed for participants in the second and highest quartiles, whose baseline FI scores were 0.022 (*β* = 0.022, 95% CI = 0.007–0.037, *p* = 0.004) and 0.026 (*β* = 0.026, 95% CI = 0.010–0.041, *p* = 0.001) units higher, respectively, compared with those in the lowest quartile (Quartile 1). 

### 3.2. UPF Consumption Frequency and FI Progression over Time

Linear mixed-effects models were used to examine the longitudinal associations between the baseline UPF consumption frequency and changes in FI score over time. A higher UPF consumption frequency at baseline was significantly associated with a greater FI score across a median follow-up of 16.14 years (Table 3).

No significant interactions between UPF consumption and time were observed, which suggests that the rates of change in the FI score did not differ significantly across UPF consumption quartiles. Consistent with the cross-sectional findings, older adults with more frequent UPF consumption, particularly those in the second (*β* = 0.015, 95% CI = 0.004–0.030, *p* = 0.045) and highest (*β* = 0.022, 95% CI = 0.006–0.037, *p* = 0.006) quartiles, experienced greater increases in frailty over time. Additionally, being female, older at baseline, and having a higher BMI were associated with FI progression, while having higher educational achievements and being enrolled in the Bagno a Ripoli study site were inversely associated with changes in FI score over time.

Taken together, the cross-sectional and longitudinal findings underscore the potential long-term contribution of frequent UPF consumption to overall poorer health status and increased vulnerability in older adults.

## 4. Discussion

This study investigated both the cross-sectional and longitudinal associations between baseline UPF consumption frequency and frailty, measured using the FI, in the InCHIANTI study. After a median of 16.14 years of follow-up, we found that older adults who consumed UPFs more frequently had the highest baseline FI score, and this difference persisted over the follow-up period compared to those with lower consumption frequency. These findings provided further evidence of the negative health impacts associated with UPFs, suggesting that more frequent UPF consumption not only impacts the frailty status in older adults cross-sectionally but is also associated with future frailty burden.

Frailty is a complex and multidimensional condition that reflects a reduction in the physiological reserves and functional capacity of older adults. Previous investigations have focused on the pro-inflammatory and anti-inflammatory components of individuals’ diet to elucidate the role of nutrition in frailty [26,38]. UPFs generally contain high levels of saturated fats and added sugars, as well as food additives such as artificial sweeteners and emulsifiers, representing the major sources of pro-inflammatory components in individuals’ diets. Multiple cohort studies conducted in different settings have demonstrated the associations between high UPF consumption and increased risk of frailty [24,26,38]. One proposed mechanism mentioned by these previous studies to explain the observed positive association between UPF consumption and frailty is increased oxidative stress due to the imbalanced consumption of anti-inflammatory and pro-inflammatory foods.

From a nutritional aspect, more frequent intake of UPFs is associated with higher intakes of total energy [39], refined sugar, and various types of fats (including the poor-quality fat resulting from excessive industrial processes) [40]. A direct consequence of surplus intake of energy, refined sugar, and fat is an accumulation of fat mass in the body, which can contribute to overweight and obesity in the long term. Obesity, a known risk factor for frailty [41,42], promotes a cascade of physiological changes, including oxidative stress, systematic inflammation, insulin resistance, and metabolic dysregulation [43], all of which are known risk factors for other chronic diseases and can negatively impact muscle functions and trophism [44]. On the other hand, consuming large amounts of UPFs may replace the consumption of minimally processed foods that contain nutrients that are important to maintain immune system function. When considering the relationships between the consumption of UPFs and the consumption of unprocessed foods, Martini et al. found the largest decrease to be in the consumption of poultry, red meat, cereals, and fruits as UPF consumption increases [18]. For poultry and red meat specifically, the estimated variation between the low UPF (15% of total daily energy share) and high UPF (75% of total daily energy share) diets roughly corresponded to a substantial decrease from one serving per day to two servings per week [18]. A comparable reduction in the consumption of cereals and legumes, approximating one serving per week, was also observed [18]. Cereals and legumes, along with vegetables and fruits, constitute primary sources of dietary fiber. Fiber is recognized as a critical factor in reducing diet-related inflammation [45], owing to its role in modulating the composition and diversity of the gut microbiome [46,47] and promoting the production of short-chain fatty acids that regulate lipid and glucose metabolisms via activating the secretion of hormones such as glucagon-like peptide 1 (GLP-1) [48] and peroxisome proliferator-activated receptor (PPAR-γ) [49]. While few studies have investigated the direct associations between dietary fiber intake and frailty, results from the Baltimore Longitudinal Study of Aging suggested that a diet rich in fiber relative to its proportion of carbohydrates was associated with lower FI scores and slower progression of the FI over time [50].

From a non-nutritive aspect, food additives used in UPFs to improve or maintain their taste and freshness may act as the hidden driver of low-grade inflammation [51]. Among the most studied are the non-caloric sweeteners, such as sucralose. For example, consumption of sucralose has been demonstrated to exacerbate the expression of bacterial pro-inflammatory mediators such as lipopolysaccharide (LPS) [52]. LPS can activate various pathways, particularly the NK-κB pathway, leading to elevated levels of pro-inflammatory mediators like interleukin 6 and 8 (IL-6, IL-8) in the body [53]. The increased levels of these pro-inflammatory mediators can induce and exacerbate age-related loss of muscle mass and function [54]. Future studies should incorporate biomarker data to further elucidate the mechanisms that link UPF with frailty in older adults.

Our study has several strengths. First, we examined the association between overall consumption frequency of various UPFs and frailty risk rather than focusing on individual food groups. This approach allows for the consideration of potential cumulative and synergistic effects of diverse foods and nutrients on frailty development and progression. In addition, the prospective design of the InCHIANTI study and the repeated measures of participants’ frailty status at follow-up visits provide us with the opportunity to identify FI trajectory patterns over time and investigate the longitudinal association between UPF consumption and frailty progression over time. Other advantages included the use of a validated FFQ to obtain dietary intake information and the use of the FI that considers multiple health deficits to evaluate frailty status.

Despite these strengths, our study is not without limitations. The baseline dietary intake assessment provided only a single time point of participants’ UPF consumption status, which failed to capture the potential change in UPF consumption during the follow-up period. Due to limited availability of data on the energy content of UPFs within the InCHIANTI study, calculating the percentage of total energy intake attributable to UPF consumption was not feasible. Additionally, the lack of a standardized definition for food ultra-processing will lead to potential misclassification of the food groups. To address this issue, we implemented a rigorous four-step process to categorize food items that included expert consultation for foods where consensus on categorization was not reached. Lastly, while we adjusted for potential confounders in our analysis, residual confounding still cannot be eliminated due to the observational nature of this study.

## 5. Conclusions

In conclusion, a higher UPF consumption frequency is associated with a higher FI score, and these differences persisted through the follow-up period. The findings of this study highlight the potential adverse impact of frequent UPF consumption on healthy aging and underscore the importance of public health strategies aimed at reducing UPF intake while promoting more nutritious dietary choices to support a longer health span. However, it is important to note that food misclassification may occur due to the current lack of a standardized definition for UPFs. At the same time, food processing is a common and often necessary practice to ensure food safety and edibility. Not all processed foods warrant restriction to promote overall well-being. These complexities highlight the pressing need for a consistent and practical definition of UPF, which is essential for developing clearer nutritional guidelines for older adults and for advancing research on broader health impacts of UPF consumption.

## Figures and Tables

**Table 1 geriatrics-10-00123-t001:** Baseline sociodemographic characteristics of InCHIANTI study participants aged 65 years and older (mean (standard deviation) or n (percentage)) by ultra-processed foods (UPFs) consumption frequency quartiles.

Variable	Total			UPF		*p*- Value ^1^
		Quartile 1 (Lowest)	Quartile 2	Quartile 3	Quartile 4	
n	938	230	240	248	220	
Age (years)	74.0 (6.6)	72.9 (6.0)	74.2 (6.8)	74.7 (6.7)	74.2 (6.8)	0.023
Female (%)	518 (55.2)	120 (52.2)	110 (45.8)	99 (39.9)	91 (41.4)	0.035
BMI (kg/m^2^)	27.5 (4.1)	27.5 (3.9)	27.6 (4.2)	27.5 (4.1)	27.5 (4.2)	0.98
Energy Intake (kcal/day)	1936.1 (552.0)	1973 (583.0)	1867 (517.0)	1853 (528.0)	2067 (558.0)	<0.001
Smoking (%)						
*Currently*	133 (14.2)	33 (14.3)	34 (14.2)	34 (13.7)	32 (14.5)	0.970
*Previously*	256 (27.3)	68 (29.6)	61 (25.4)	66 (26.6)	61 (27.7)	
*Never*	549 (58.5)	129 (56.1)	145 (60.4)	148 (59.7)	127 (57.7)	
Education (years)	5.5 (3.2)	6.0 (3.8)	5.5 (3.4)	5.4 (3.0)	5.1 (2.6)	0.027
Study site (%)						
*Greve in Chianti*	445 (47.4)	115 (50.0)	112 (46.7)	121 (48.8)	97 (44.1)	0.608
*Bagno a Ripoli*	493 (52.6)	115 (50.0)	128 (53.3)	127 (51.2)	123 (55.9)	
Baseline Frailty Index (FI)	0.133 (0.10)	0.108 (0.08)	0.140 (0.10)	0.138 (0.10)	0.146 (0.10)	<0.001

^1^ *p*-value was calculated from ANOVA and Chi-Square tests that compared the differences in each characteristic across UPF frequency quartiles.

**Table 2 geriatrics-10-00123-t002:** Cross-sectional association between ultra-processed foods (UPFs) consumption frequency and FI at baseline of the InCHIANTI study participants aged 65 years and older, adjusted for sociodemographic characteristics.

Variables	*β*s (Estimates)	95% CI	*p*-Value
UPF consumption frequency			
*Quartile 1*		(Ref)	
*Quartile 2*	0.022	(0.007, 0.037)	0.004
*Quartile 3*	0.014	(−0.001, 0.029)	0.071
*Quartile 4*	0.026	(0.010, 0.041)	0.001
Baseline age (years)	0.007	(0.006, 0.007)	<0.001
Sex (females)	0.025	(0.012, 0.038)	0.0002
BMI (kg/m^2^)	0.003	(0.001, 0.004)	<0.001
Smoking			
*Previously*	−0.008	(−0.026, 0.010)	0.369
*Never*	−0.018	(−0.035, −0.0007)	0.041
Education (years)	−0.003	(−0.004, −0.0008)	0.004
Study site			
*Bagno a Ripoli*	−0.015	(−0.026, −0.005)	0.005

The referent group of “sex” in the model was “males”. The referent group of “smoking” status in the model was “currently smoking”. The referent group of “study site” in the model was “Greve in Chianti”.

**Table 3 geriatrics-10-00123-t003:** Longitudinal associations between ultra-processed foods (UPFs) consumption frequencies and FI progression over all visits of InCHIANTI study participants aged 65 years and older, adjusted for sociodemographic characteristics.

Variables	*β*s (Estimates)	95% CI	*p*-Value
UPF consumption frequency			
*Quartile 1*		(Ref)	
*Quartile 2*	0.015	(0.004, 0.030)	0.045
*Quartile 3*	0.010	(−0.005, 0.025)	0.197
*Quartile 4*	0.022	(0.006, 0.037)	0.006
**Follow-up years** (by visit)	0.012	(0.011, 0.013)	<0.001
Baseline age (years)	0.007	(0.007, 0.008)	<0.001
Sex (females)	0.024	(0.012, 0.037)	0.0002
BMI (kg/m^2^)	0.003	(0.001, 0.004)	<0.001
Smoking			
*Previously*	−0.004	(−0.021, 0.013)	0.655
*Never*	−0.011	(−0.028, 0.006)	0.193
Education (years)	−0.003	(−0.005, −0.001)	0.001
Study Site			
*Bagno a Ripoli*	−0.015	(−0.026, −0.005)	0.005

FI data included in this longitudinal analysis were collected from baseline to visit three. The referent group of “sex” in the model was “males”. The referent group of “smoking” status in the model was “currently smoking”. The referent group of “study site” in the model was “Greve in Chianti”.

## Data Availability

The data presented in this study are available upon request from the corresponding author (S.T.).

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
