# Peer review of "Association Between Ultra-Processed Food Consumption Frequency and Frailty: Findings from the InCHIANTI Study of Aging"

_geriatrics, 2025, doi:10.3390/geriatrics10050123_

Round 1

Reviewer 1 Report

Comments and Suggestions for Authors

Diet and healthy aging have become important health issues lately. So the topic is of great interest.

My comments are the following:

  • the results should be presented more detailed. I suggest not to begin the “Results“ section with a table.
  • in table 1 - education (measured in years) is between 5.1 and 5.5 years. Are these values real? 
  • I suggest to redesign both table 2 and 3 as they can be hardly understood 
  • Also I would suggest to better describe the results focusing your attention on explaining the statistical results, not just presenting them.

Author Response

Thank you for all your comments and suggestions!

Comment 1: The results should be presented more detailed. I suggest not to begin the “Results“ section with a table.

Response 1: Thank you for your comment and the suggestion. We have restructured the “Results” section and have expanded the description to provide more relevant information about the findings. Please see Lines 177-245 for the updates.

Comment 2: In Table 1 - education (measured in years) is between 5.1 and 5.5 years. Are these values real? 

Response 2: Thank you for your comment. We have carefully reviewed our coding and consulted experts who are familiar with the demographics of the InCHIANTI population1-3. We confirm that the reported average years of education for InCHIANTI participants are accurate. The InCHIANTI study participant recruitment started in 1998 and primarily included Italian older adults residing in Tuscany, Italy. In this region of Italy, particularly for this older age group at baseline visit, it was common for people to have very low years of education (primary education only) and join the workforce. Thus, the average reported experience of the target population.

Comment 3: I suggest to redesign both table 2 and 3 as they can be hardly understood 

Response 3: Thank you for your comment and suggestion. We have redesigned both Tables 2 and 3 to improve clarity and readability. Please see the updates in the revised Result section (Lines 177-245).

Comment 4: Also I would suggest to better describe the results focusing your attention on explaining the statistical results, not just presenting them.

Response 4: Thank you for your comment and recommendation. We agree that providing more explanation of the statistical findings is critical for proper interpretation. We have revised the corresponding paragraphs in the Result section to expand the description and our interpretation of the results presented in the tables. Please see the revised Result section (Lines 177-245) for the updates.  

References

  1. Stringa N, van Schoor NM, Milaneschi Y, et al. Physical Activity as Moderator of the Association Between APOE and Cognitive Decline in Older Adults: Results from Three Longitudinal Cohort Studies. J Gerontol A Biol Sci Med Sci. Sep 25 2020;75(10):1880-1886. doi:10.1093/gerona/glaa054
  2. Hoogendijk EO, Stenholm S, Ferrucci L, Bandinelli S, Inzitari M, Cesari M. Operationalization of a frailty index among older adults in the InCHIANTI study: predictive ability for all-cause and cardiovascular disease mortality. Aging Clin Exp Res. Jun 2020;32(6):1025-1034. doi:10.1007/s40520-020-01478-3
  3. Cesari M, Pahor M, Bartali B, et al. Antioxidants and physical performance in elderly persons: the Invecchiare in Chianti (InCHIANTI) study. Am J Clin Nutr. Feb 2004;79(2):289-94. doi:10.1093/ajcn/79.2.289

Reviewer 2 Report

Comments and Suggestions for Authors

General comment

Thank you very much for the opportunity to review this article. This study demonstrates the relationship between ultra-processed foods and frailty. Overall, it is written in an easy-to-understand manner, and there were no inconsistencies in the methods, results, or discussion.

Minor comments

  • Isn't the first author name for reference number 17 “Cardoso BR”?
  • The title of the article for reference number 35 is written in all capital letters, the title should be corrected to match the referenced article.

Author Response

Thank you for all your comments! 

Comment 1: Isn't the first author name for reference number 17 “Cardoso BR”?

Response 1: Thank you for pointing this out. We have checked the original article and confirmed the correct author information. Reference number 17 has now been updated accordingly.

Comment 2: The title of the article for reference number 35 is written in all capital letters, the title should be corrected to match the referenced article.

Response 2:  Thank you for your comment. We have checked the original article and found that its title was written in all capital letters. But for consistency with the formatting of the other references, we have updated this title accordingly.

Round 2

Reviewer 1 Report

Comments and Suggestions for Authors

The manuscript can be accepted in the current form.